# A Toy Model of the Information Paradox in Empty Space

Suvrat Raju

*International Centre for Theoretical Sciences, Tata Institute of Fundamental Research, Shivakote, Bengaluru 560089, India.*

A sharp version of the information paradox involves a seeming violation of the monogamy of entanglement during black hole evaporation. We construct an analogous paradox in empty anti-de Sitter space. In a local quantum field theory, Bell correlations between operators localized in mutually spacelike regions are monogamous. We show, through a controlled calculation, that this property can be violated by an order-1 factor in a theory of gravity. This example demonstrates that what appears to be a violation of the monogamy of entanglement may just be a subtle violation of locality in quantum gravity.

## INTRODUCTION

In this paper, we present a toy model that captures key aspects of the information paradox in a setting that facilitates clean calculations. In a local quantum field theory, Bell correlations between different spatial regions are monogamous. We show that this monogamy is violated dramatically in a theory of quantum gravity, even in empty anti-de Sitter space. This produces a "paradox" that is analogous to the "cloning" and "monogamy" paradoxes for evaporating black-holes. This construction provides strong evidence that these paradoxes can be resolved by recognizing that degrees of freedom cannot be localized in quantum gravity and information that appears to be present in one region of space can also be extracted from another region.

The cloning paradox arises because it is possible to draw nice slices in the evaporating black hole spacetime so that a single spacelike slice intersects both the infalling matter and captures a large fraction of the outgoing Hawking radiation. This makes it appear that the same information is present at two points on the slice, violating no-cloning theorems in quantum mechanics. A related, and sharper paradox was constructed in [1] and elaborated in [2]. A consideration of the Hawking process reveals that, for an old black hole, the near-horizon region must be entangled with the interior of the black-hole and also with the early Hawking radiation that may have traveled far from the horizon. This appears to violate information-theoretic inequalities on the monogamy of entanglement and again suggests that information in the interior has been "cloned" in the exterior.

The papers [3] proposed a resolution to these paradoxes relying on the idea that, in quantum gravity, degrees of freedom in one region can sometimes be equated to a combination of degrees of freedom in another region. The existence of this physical effect, called "complementarity" [4], was demonstrated in a simple setting in [5] and this paper will elucidate its relation to the information paradox.

A complete understanding of black-hole evaporation requires additional physical effects. For instance, nonperturbative dynamical effects in gravity provide the exponentially small corrections that are required to unitarize Hawking radiation and reconcile the late-time behaviour of two-point functions or the spectral form-factor with general predictions from unitarity [6–9]. Moreover, to resolve paradoxes that appear in the interior of large AdS black-holes [10], the map between the bulk and boundary must be state-dependent [11].

However these three effects — complementarity, exponentially small corrections and state-dependence are really independent physical effects in gravity and should not be conflated. In this paper, we will explore complementarity but we will *not* appeal either to state-dependence or to non-perturbative corrections.

A technical point emphasized in this paper is that the monogamy of entanglement in gravity is best studied by examining the monogamy of Bell correlations. Monogamy paradoxes for black hole evaporation were originally formulated in terms of the strong subadditivity of the von Neumann entropy. However, the von Neumann entropy is difficult to even define [12], let alone compute, in a theory of dynamical gravity. In contrast, Bell correlations can be computed reliably in perturbation theory as we show here.

There are no gauge-invariant exactly local operators in gravity [13]. In this paper, we will use the term *localized* to describe operators that comprise quantum fields from a region after a specific choice of gauge. The reader should keep this definition of the term *localized* in mind to avoid any confusion with other notions of localization. Except at one point below, we leave the gauge-choice unspecified since different choices of gauge just change the results by $O\left(\frac{1}{N}\right)$.

A localized operator, as we have defined it, is *not* an exactly local operator. However, localized operators are often used (see, for example [14]) to capture naive physical notions of a local observation. Most recent versions of the information paradox tacitly assume that, at low energies and up to leading order in $\frac{1}{N}$, such localized observables will share the properties of exactly local operators. For instance, essential to the paradox of [1, 2] is the idea that because the old Hawking radiation has low energy and can be manipulated by localized operators far away from the black hole, such manipulations do

not affect the black-hole interior.

The result of this paper show, in a precise setting, that such an assumption is wrong. If one allows complicated enough operations with localized operators, then it may not be true that the effect of these operations remains confined to the original region, and such a nonlocal effect may be important at $O(1)$. For complicated localized operators, not only can their commutator become large at spacelike separation but, crucially, "complementarity" can emerge so that the same quantum information is available in operators localized in distinct regions. We demonstrate this explicitly in this paper by using complicated localized operators to construct an analogy to the cloning and monogamy paradoxes even in empty space.

We will work with a minimally coupled scalar field, $\phi$ in anti-de Sitter space, with $\ell_{\mathrm{AdS}} = 1$. The Planck length, in these units, is denoted by $\frac{1}{N}$; the same parameter is assumed to control the self-interactions of the field.

## MONOGAMY OF ENTANGLEMENT

We start by reviewing how the monogamy of entanglement constrains Bell correlations.

Consider two pairs of operators $\{A_1, A_2\}$ and $\{B_1, B_2\}$ with operator norms, $\|A_i\|, \|B_i\| \le 1$ and with $[A_i, B_j] = 0$. Define the *CHSH* operator [15]

$$\mathcal{C}_{AB} = A_1(B_1 + B_2) + A_2(B_1 - B_2). \qquad (1)$$

Classically, in any state, $|\langle \mathcal{C}_{AB} \rangle| \le 2$. But quantum mechanically, $|\langle \mathcal{C}_{AB} \rangle| \le 2\sqrt{2}$ [16]. Thus a state that yields $2 < |\langle \mathcal{C}_{AB} \rangle| \le 2\sqrt{2}$ displays correlations between the operators $A_i$ and $B_i$ beyond classical correlations — which implies that the degrees of freedom probed by these operators are entangled.

A beautiful statement of the monogamy of entanglement is then as follows [17]. (See also [18].) Consider a *third* pair of operators $\{C_1, C_2\}$ with $\|C_i\| \le 1, [A_i, C_j] = [B_i, C_j] = 0$ and form the combination $\mathcal{C}_{AC}$ just as above. Then in any state we have

$$\langle \mathcal{C}_{AB} \rangle^2 + \langle \mathcal{C}_{AC} \rangle^2 \le 8. \qquad (2)$$

Therefore if the operators $A_i$ are "entangled" with $B_i$ then their correlations with $C_i$ must be less than the allowed *classical* limit: $|\langle \mathcal{C}_{AB} \rangle| > 2 \implies |\langle \mathcal{C}_{AC} \rangle| < 2$.

In a local quantum field theory, the criterion that $A_i, B_i, C_i$ commute can be replaced by the statement that $A_i, B_i, C_i$ are localized on spacelike separated regions.

We will now show that in a theory of gravity, if we consider operators $A_i, B_i, C_i$ localized on spacelike separated regions, then (2) is violated by an $O(1)$ amount.

## BELL INEQUALITIES IN FIELD THEORY

Much of the literature on Bell inequalities is focused on qubits, and while Bell inequalities have been considered in quantum field theory [19–21], here we will independently construct some simple operators that display Bell correlations beyond the classical limit.

### Preliminaries

To warm up, consider a system of two commuting simple harmonic oscillators, with annihilation operators $\alpha_s$, where $s = A$ or $s = B$, in a thermofield state

$$\sqrt{1 - x^2} e^{x\alpha_A^\dagger \alpha_B^\dagger} |0\rangle. \qquad (3)$$

Here $|0\rangle$ is the joint vacuum and $0 < x < 1$.

Let $P_s$ be the projector onto states annihilated by $\alpha_s$. Take the CHSH operator (1) to comprise

$$A_1 = P_A - \alpha_A^\dagger P_A \alpha_A; \quad A_2 = \alpha_A^\dagger P_A + P_A \alpha_A;$$
$$B_1 = \frac{1}{\sqrt{2}} \left( P_B - \alpha_B^\dagger P_B \alpha_B + \alpha_B^\dagger P_B + P_B \alpha_B \right); \qquad (4)$$
$$B_2 = \frac{1}{\sqrt{2}} \left( P_B - \alpha_B^\dagger P_B \alpha_B - \alpha_B^\dagger P_B - P_B \alpha_B \right).$$

It can be easily checked that $\|A_i\| = \|B_i\| = 1$. In the thermofield state

$$\langle \mathcal{C}_{AB} \rangle = \sqrt{2}(1 - x)(1 + x)^3 \qquad (5)$$

This is maximized at $x = 1/2$ with $\langle \mathcal{C}_{AB} \rangle = \frac{27\sqrt{2}}{16} \approx 2.4$. This does not saturate the bound, $|\langle \mathcal{C}_{AB} \rangle| \le 2\sqrt{2}$, but will suffice for our purpose.

### Bell operators in quantum field theory

We now turn to a weakly interacting quantum field theory. All expectation values below will be taken in the vacuum $|\Omega\rangle$. The idea is to extract a pair of simple harmonic degrees of freedom for which the vacuum resembles the thermofield state (3).

By smearing the field and its conjugate momentum with functions supported on spatially separated compact regions, we define two Hermitian operators $(X_s, \Pi_s)$ and set $\alpha_s = \frac{1}{\sqrt{2}}(X_s + i\Pi_s)$. Here, as above, $s$ runs over systems "A" and "B" and the smearing functions are normalized by $[\alpha_s, \alpha_{s'}^\dagger] = \delta_{ss'}$.

The projector onto states annihilated by $\alpha_s$ is

$$P_s = \frac{1}{\pi^2} \int_{-\infty}^{\infty} d^2\vec{t} \int_0^{2\pi} d\theta_s \frac{e^{-\vec{t}^2 - \kappa(\theta_s)(t_1 X_s - t_2 \Pi_s)}}{(e^{i\theta_s} - 1)}, \qquad (6)$$

where $\vec{t} = (t_1, t_2)$ is a two-component vector of dummy variables and $\kappa(\theta) \equiv 2\sqrt{\tanh(i\theta)}$. Formula (6) can be

verified by performing the integrals over the dummy variables carefully, using the BCH lemma to account for the non-commutativity of $X_s$ and $\Pi_s$

Using these projectors we define operators in quantum field theory precisely as in (4). The most general two-point function of these operators can be extracted from

$$\mathcal{Q}[v_i, \zeta_i] = \int \frac{d^2\vec{t}\, d^2\vec{y}\, d\theta_A d\theta_B}{\pi^4(e^{i\theta_B}-1)(e^{i\theta_A}-1)} e^{-\vec{t}^2-\vec{y}^2}\langle\mathcal{G}\rangle, \quad (7)$$

with

$$\mathcal{G} = e^{v_2\alpha_B^\dagger} e^{X_B\tilde{y}_1-\Pi_B\tilde{y}_2} e^{\zeta_2\alpha_B} e^{v_1\alpha_A^\dagger} e^{\tilde{t}_1 X_A-\Pi_A\tilde{t}_2} e^{\zeta_1\alpha_A},$$

and $\tilde{t}_i = t_i\kappa(\theta_A)$; $\tilde{y}_i = y_i\kappa(\theta_B)$. The values of $\mathcal{Q}$ and its derivatives at $v_i = \zeta_i = 0$ yield correlators of all operators in (4).

For actual computations, it is convenient to express $\alpha_s$ in terms of global creation and annihilation operators, labeled by quantum numbers $n$ and $\ell$, that satisfy $[a_{n,\ell}, a_{n',\ell'}^\dagger] = \delta_{nn'}\delta_{\ell\ell'}$. (Such operators can even be found in the interacting theory.)

$$\alpha_s = \sum_{n,\ell} h_s(n,\ell)a_{n,\ell} + g_s^*(n,\ell)a_{n,\ell}^\dagger.$$

Since $[\alpha_s, \alpha_{s'}^\dagger] = \delta_{ss'}$,

$$h_s \cdot h_{s'}^* - g_s^* \cdot g_{s'} = \delta_{ss'},$$

where the dot-product is taken by summing over $n, \ell$: $h_s \cdot h_{s'}^* \equiv \sum h_s(n,\ell)h_{s'}^*(n,\ell)$.

Evaluating (7) for arbitrary $h_s, g_s$ is a straightforward, albeit tedious, exercise. We only outline the steps. First,

$$\langle\mathcal{G}\rangle = \exp\Big[\sum_{p,q=1}^4 (f_p\cdot f_q^* + f_q\cdot f_p^*)\frac{m_p m_q}{4} - \frac{\mathcal{R}}{2}\Big] + O\left(\frac{1}{N}\right),$$

where $f_1 = (h_A + g_A)$; $f_2 = -i(h_A - g_A)$; $f_3 = h_B + g_B$; $f_4 = -i(h_B - g_B)$; $\zeta_i^\pm = (\zeta_i \pm v_i)/\sqrt{2}$; $m_1 = \tilde{t}_1 + \zeta_1^+$, $m_2 = -\tilde{t}_2 + i\zeta_1^-$; $m_3 = \tilde{y}_1 + \zeta_2^+$; $m_4 = -\tilde{y}_2 + i\zeta_2^-$ and

$$\mathcal{R} = \left(m_1\zeta_1^+ + im_2\zeta_1^- + m_3\zeta_2^+ + im_4\zeta_2^-\right) - \zeta_1 v_1 - \zeta_2 v_2.$$

The $O\left(\frac{1}{N}\right)$ corrections above arise because, in an interacting theory, the vacuum is not exactly annihilated by the global annihilation operators.

The remaining integrals in (7) over $\vec{t}$ and $\vec{y}$ are Gaussian. They yield a function with a regular Fourier series expansion in $\theta_A$ and $\theta_B$ whose zeroth order term is the desired answer. The final expression for arbitrary $h_s, g_s$ is unenlightening; so we do not record it here.

## Entangled modes in AdS

The discussion above applies to any quantum field theory but we now turn to global AdS and make specific

choices of $h_s$ and $g_s$ to obtain simple answers for Bell correlations.

The global AdS metric is

$$ds^2 = \frac{1}{\cos^2\rho}\left(-dt^2 + d\rho^2 + \sin^2\rho\, d\Omega_{d-1}^2\right). \quad (8)$$

A minimally coupled massive scalar dual to an operator of dimension $\Delta$ can be expanded as

$$\phi(t,\rho,\Omega) = \sum_n a_{n,\ell}e^{-i(2n+\ell+\Delta)t}Y_\ell(\Omega)\chi_{n,\ell}(\rho) + \text{h.c.},$$

up to $O\left(\frac{1}{N}\right)$, where $Y_\ell$ are spherical harmonics. The wave-functions $\chi_{n,\ell}$ are given in [5] but we will only need their asymptotic forms. We set the normalization so that $[a_{n,\ell}, a_{n',\ell'}^\dagger] = \delta_{nn'}\delta_{\ell\ell'}$.

Now consider the $(d-1)$-sphere in AdS at $\rho = \rho_0$ and $t = 0$. We will construct the operators $A_i$ by smearing the field slightly inside the contracting light shell from this sphere, and $B_i$ by smearing slightly outside the expanding light shell.

To be precise, we consider a real-valued "turning on/off" function $\mathcal{T}(U)$ that is largely constant in an interval $U_l \leq U \leq U_h$, and vanishes smoothly at these endpoints. We take the limit where $U_0 \to 0$, $\log(U_l/U_0) \to -\infty$, $\log(U_h/U_0) \to \infty$ but yet $U_h \ll 1$. These cutoffs are introduced to make all integrals below convergent, and we will denote any dependence on them by the symbol $O(\epsilon)$. The cutoffs never scale with $N$ and so $O\left(\frac{1}{N}\right) \ll O(\epsilon)$.

We define $\tilde{\mathcal{T}}(\nu)$, which is sharply centered around a particular frequency, $\omega_0$, by

$$\mathcal{T}(U)\left(\frac{U}{U_0}\right)^{i\omega_0} = \int \tilde{\mathcal{T}}(\nu)\left(\frac{U}{U_0}\right)^{i\nu} d\nu.$$

With some prescience, we also impose

$$\lim_{\nu\to 0}\frac{1}{\nu}\tilde{\mathcal{T}}(\nu) = 0; \quad \pi\int|\tilde{\mathcal{T}}(\nu)|^2\frac{d\nu}{\nu} = 1$$

With $\rho_A(U) \equiv \rho_0 - v_0 - U/2$ and $t_A(U) \equiv U/2 - v_0$, and $\rho_B(U) \equiv \rho_0 + v_0 + U/2$; $t_B(U) \equiv v_0 - U/2$, where $v_0$ is an irrelevant small positive constant, we take

$$\alpha_s = \int \frac{dU}{U}d^{d-1}\Omega\Big[\phi(\rho_s(U), t_s(U), \Omega) \\ \times [\tan\rho_s(U)]^{\frac{d-1}{2}}\left(\frac{U}{U_0}\right)^{i\omega_s}\mathcal{T}(U)\Big], \quad (9)$$

where $\omega_A = \omega_0$ and $\omega_B = -\omega_0$. In the limit of interest the two modes are defined by integrals that have effectively vanishing support in the global AdS geometry but nevertheless $(U/U_0)^{i\omega_s}$ undergoes a large number of oscillations in this region.

The functions $h_s(n,\ell)$ and $g_s(n,\ell)$ vanish for $\ell \neq 0$ (because of the $\Omega$ integral) and are effectively supported

only by large values of $n$. The AdS wave-function effectively remains constant in the integration region for modes with $\mathrm{O}(1)$ values of $n$ and the integrals defined by (9) then vanish since $\tilde{\mathcal{T}}(\nu)$ vanishes for small $\nu$.

For large $n$, the radial wave-functions simplify greatly.

$$\chi_{n,\ell}(\rho) \underset{n\to\infty}{\longrightarrow} \frac{1}{\sqrt{\pi n}} \cot(\rho)^{\frac{d-1}{2}} \sin\left(\xi_0 - \rho(\Delta + \ell + 2n)\right).$$

where $\xi_0 = \frac{\pi}{4}(d+1+2\ell+4n)$. Then, for $n \gg 1$, neglecting $\mathrm{O}(\epsilon)$-terms,

$$h_A(n,0) = \frac{e^{-i\xi_1}}{2\sqrt{\pi n}} \int e^{\frac{\pi\nu}{2}} (2U_0 n)^{-i\nu} \Gamma(i\nu)\tilde{\mathcal{T}}(\nu)d\nu;$$

$$g_A^*(n,0) = \frac{e^{i\xi_1}}{2\sqrt{\pi n}} \int e^{-\frac{\pi\nu}{2}} (2U_0 n)^{-i\nu} \Gamma(i\nu)\tilde{\mathcal{T}}(\nu)d\nu;$$

$$h_B(n,0) = \frac{e^{-i\xi_1}}{2\sqrt{\pi n}} \int e^{\frac{\pi\nu}{2}} (2U_0 n)^{i\nu} \Gamma(-i\nu)\tilde{\mathcal{T}}^*(\nu)d\nu;$$

$$g_B^*(n,0) = \frac{e^{i\xi_1}}{2\sqrt{\pi n}} \int e^{\frac{-\pi\nu}{2}} (2U_0 n)^{i\nu} \Gamma(-i\nu)\tilde{\mathcal{T}}^*(\nu)d\nu,$$

where $\xi_1 = \xi_0 + (\Delta + 2n)\rho_0 - (d+1)\frac{\pi}{2}$.

To sum products of these functions over $n$, we recognize that in the limit of interest,

$$\sum \frac{1}{n} (U_0 n)^{it} \to \int d\log(U_0 n) e^{it\log(U_0 n)} = 2\pi\delta(t) + \mathrm{O}(\epsilon).$$

Thus, for example, up to $\mathrm{O}(\epsilon)$,

$$h_A \cdot h_A^* = \int d\nu \frac{\pi e^{\pi\nu}}{2\nu \sinh(\pi\nu)} |\tilde{\mathcal{T}}(\nu)|^2 d\nu = \frac{e^{\pi\omega_0}}{2\sinh(\pi\omega_0)}.$$

Proceeding as above, with $x = e^{-\pi\omega_0}$,

$$f_p \cdot f_q^* + f_q \cdot f_p^* = \frac{2}{1-x^2} \begin{pmatrix} x^2+1 & 0 & 2x & 0 \\ 0 & x^2+1 & 0 & -2x \\ 2x & 0 & x^2+1 & 0 \\ 0 & -2x & 0 & x^2+1 \end{pmatrix}$$

Substituting this into the integral (7), we obtain *precisely* the answer (5). In particular, for $x = 1/2$

$$\langle \mathcal{C}_{AB} \rangle = \frac{27\sqrt{2}}{16} + \mathrm{O}\left(\frac{1}{N}\right) + \mathrm{O}(\epsilon).$$

## A PARADOX IN GRAVITY

We now turn to the effect of gravity. We will construct operators $C_i$ by smearing the field on a region spatially separated from the regions used for $A_i$ and $B_i$.

Our construction relies on the fact that in a theory of gravity (and only in a theory of gravity!), we can construct a bulk operator, near the boundary of the space, that projects onto the vacuum. The simplest way to define this operator is to expand the metric as

$$g_{\mu\nu} = g_{\mu\nu}^{\mathrm{AdS}} + h_{\mu\nu},$$

where $g_{\mu\nu}^{\mathrm{AdS}}$ is given in (8) and choose Fefferman-Graham gauge, $h_{\rho\mu} = 0$, near the boundary. In the quantum theory, $h_{\mu\nu}$ is an *operator*, and we now consider

$$H^{\mathrm{can}} = \lim_{\rho\to\frac{\pi}{2}} (\cos\rho)^{2-d} \int d^{d-1}\Omega \, \frac{h_{tt}}{16\pi G_N},$$

where the integral is along $t = 0$. By the standard "extrapolate" dictionary [22] in AdS/CFT [23–25], $H^{\mathrm{can}}$ is the Hamiltonian. $H^{\mathrm{can}}$ may also be expressed covariantly [26] and the fact that it is a boundary term is a well known fact in gravity [27].

Even if the extrapolate dictionary is corrected at $\mathrm{O}\left(\frac{1}{N}\right)$, here we will only need the following: $H^{can}$ *is a positive operator and has a unique eigenstate with eigenvalue zero whose overlap with the ground state of the full theory is* $1 - \mathrm{O}\left(\frac{1}{N}\right)$. Note we are in global AdS, where the vacuum is unique and the spectrum is gapped; this avoids any difficulties with infrared modes.

The assumption above is also physically well motivated. In fact, in our own Universe, we measure the mass of distant objects by measuring the falloff in their gravitational fields, and quantum gravity should not allow a positive-energy object to hide its effect in the distant field.

Now consider

$$\mathcal{P}_\Omega = \lim_{z\to\infty} e^{-zH^{\mathrm{can}}}.$$

Although this operator seems very complicated, we can treat it exactly. By the assumption above,

$$\mathcal{P}_\Omega = |\Omega\rangle\langle\Omega| + \mathrm{O}\left(\frac{1}{N}\right).$$

We pause to briefly mention some properties of $\mathcal{P}_\Omega$, which also bring out its similarities to observables that are used in common versions of the information paradox. First, a correlator with an insertion of $\mathcal{P}_\Omega$ can be calculated to arbitrary precision by combining correlators with suitably many insertions of $h_{tt}$ [5]. Therefore correlators of $\mathcal{P}_\Omega$ are just *multi-point correlators* of graviton-fluctuations. For now, we just note that in this sense, an observation involving $\mathcal{P}_\Omega$ is similar to the observables of [1, 2] that involve high-point correlators of Hawking radiation. We will return below to a more precise statement of the similarities between these observables.

Second, one of the points made in [2] was that a suitably powerful observer could *operationally* distil information from Hawking radiation while remaining far away from the black hole. We note that an observer with access to multiple identically prepared systems and an external measuring apparatus can operationally "act" with $\mathcal{P}_\Omega$: such an observer simply has to measure $H^{\mathrm{can}}$ near the boundary and discard the results of experiments that yield $H^{\mathrm{can}} \neq 0$.

To project onto the vacuum from afar is possible only in gravity but the second step in our construction relies on the fact that it is possible to "lift" the vacuum to any excited state in any field theory as we now demonstrate.

Consider the $(d + 2)$-dimensional embedding space, with metric $\mathrm{diag}(-1, -1, 1, \ldots 1)$, where $\mathrm{AdS}_{d+1}$ is the hyperboloid, $\vec{X} \cdot \vec{X} = -1$. The global coordinates are

$$X_0 = \sec\rho\sin\tau; \ \ X_1 = \sec\rho\cos\tau; \ \ X_{j+1} = \tan\rho\,\Omega_j,$$

where $\sum_{j=1}^{d}\Omega_j^2 = 1$. Now consider a bulk causal wedge, dual to a boundary causal diamond, spanned by coordinates $\rho_R, t_R, u$ and a $(d-2)$-sphere, $\tilde{\Omega}_j$ [28]:

$$X_1 = \rho_R\cosh u\cosh\gamma + \tilde{\rho}_R\cosh(t_R)\sinh\gamma;$$
$$X_{d+1} = \rho_R\cosh u\sinh\gamma + \tilde{\rho}_R\cosh\gamma\cosh t_R;$$
$$X_0 = \tilde{\rho}_R\sinh(t_R); \ \ X_{j+1} = \rho_R\sinh u\tilde{\Omega}_j; \ \ \sum_{j=1}^{d-1}\tilde{\Omega}_j^2 = 1,$$

with $\tilde{\rho}_R^2 \equiv \rho_R^2 - 1$. The metric, in these coordinates, is

$$ds^2 = (1 - \rho_R^2)dt_R^2 + \frac{d\rho_R^2}{\rho_R^2 - 1} + \rho_R^2 dH_{d-1}^2,$$

where $u$ and $\tilde{\Omega}_j$ combine to form a unit hyperbolic space.

We set $\cosh\gamma > \sec\rho_0$ so that the entire wedge is *spacelike* to the regions that support $A_i$ and $B_i$. By integrating along $t_R = 0$, we can extract the wedge-annihilation operators

$$\eta_{\omega,\lambda} = \int(\phi + \frac{i}{\omega}\frac{d\phi}{dt_R})\psi_{\omega,\lambda}^*(\rho_R)L_\lambda^*(H)\frac{\rho_R^{d-1}d\rho_R}{\tilde{\rho}_R^2}d^{d-1}H,$$

where $L_\lambda$ are eigenfunctions of the hyperbolic Laplacian and the wave-functions $\psi_{\omega,\lambda}$ are given in [29]. We can find operators $\tilde{\eta}_{\omega,\lambda}$ on the wedge's complement so that, up to $\mathrm{O}\left(\frac{1}{N}\right)$, $[\eta_{\omega,\lambda}, \tilde{\eta}_{\omega',\lambda'}] = [\eta_{\omega,\lambda}^\dagger, \tilde{\eta}_{\omega',\lambda'}] = 0$, and

$$\tilde{\eta}_{\omega,\lambda}|\Omega\rangle = e^{-\pi\omega}\eta_{\omega,\lambda}^\dagger|\Omega\rangle; \quad \tilde{\eta}_{\omega,\lambda}^\dagger|\Omega\rangle = e^{\pi\omega}\eta_{\omega,\lambda}|\Omega\rangle. \quad (10)$$

This follows from the Bisognano-Wichmann theorem [30] and can be checked explicitly [31]. Moreover, $X_B, \Pi_B$ are linear combinations of $\eta_{\omega,\lambda}, \eta_{\omega,\lambda}^\dagger, \tilde{\eta}_{\omega,\lambda}, \tilde{\eta}_{\omega,\lambda}^\dagger$. So, using (6) and (10), we can find operators $Q_i$ comprising *only* $\eta_{\omega,\lambda}$ and $\eta_{\omega,\lambda}^\dagger$ that satisfy

$$Q_i|\Omega\rangle = B_i|\Omega\rangle + \mathrm{O}\left(\frac{1}{N}\right). \quad (11)$$

Define the Hermitian operators

$$C_i = \frac{\langle B_i^2\rangle\left(Q_i\mathcal{P}_\Omega + \mathcal{P}_\Omega Q_i^\dagger - \langle B_i\rangle\mathcal{P}_\Omega\right) - \langle B_i\rangle Q_i\mathcal{P}_\Omega Q_i^\dagger}{\langle B_i^2\rangle - \langle B_i\rangle^2}.$$

Since $\mathcal{P}_\Omega$ is localized in a small strip near the boundary and $Q_i$ are localized in the wedge, $C_i$ are localized in a

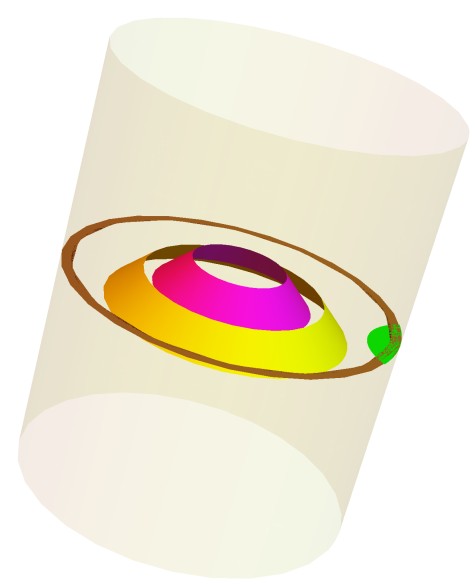

FIG. 1. *Support of the operators $A_i$ (purple), $B_i$ (yellow), $C_i$ (union of brown strip supporting $\mathcal{P}_\Omega$ and green region supporting $Q_i$) in global AdS.*

region spacelike to the regions containing $A_i$ and $B_i$. The combination above is chosen because, by (11), it satisfies

$$\|C_i\|^2 = \langle B_i^2\rangle + \mathrm{O}\left(\frac{1}{N}\right) \le 1; \ \ \langle A_jC_i\rangle = \langle A_jB_i\rangle + \mathrm{O}\left(\frac{1}{N}\right).$$

Therefore, for $x = 1/2$, $\langle\mathcal{C}_{AC}\rangle = \frac{27\sqrt{2}}{16} + \mathrm{O}\left(\frac{1}{N}\right) + \mathrm{O}(\epsilon)$. But then,

$$\langle\mathcal{C}_{AB}\rangle^2 + \langle\mathcal{C}_{AC}\rangle^2 = \frac{729}{64} + \mathrm{O}\left(\frac{1}{N}\right) + \mathrm{O}(\epsilon) \approx 11.4 > 8!$$

So we have violated the inequality (2) by an $\mathrm{O}(1)$ amount by means of operators localized in distinct spatial regions.

## CONCLUSION

The resolution to the paradox above is clear. Although the operators $B_i$ and the operators $C_i$ are localized on spacelike-separated regions, they are nevertheless secretly acting on the same degrees of freedom in the vacuum. So there is no contradiction with the quantum-information theorem (2), which assumes that operators from different pairs act on distinct Hilbert spaces.

So this construction shows that, in gravity, if one probes spacetime with fine-grained operators like $\mathcal{P}_\Omega$ then the intuitive notion that spatially separated regions contain distinct degrees of freedom may break down completely. This provides a proof of principle that for questions involving such complicated operators, even within

low-energy effective field theory, one must carefully take into account that a localized operator in one region may sometimes be equated to a combination of localized operators from another region to avoid paradoxes.

We emphasize that this is a feature of quantum gravity, and a similar construction is *not possible* in gauge theories. In gauge-theories, the charge is a boundary-term, just like gravity. However, the crucial difference is that, unlike gravity, the projector onto states of zero gauge-charge does not project onto a unique state. This would cause an attempt to repeat the construction above in gauge theories to fail.

From a technical perspective, the operators $Q_i$ are delicately tuned to use the local-entanglement in the vacuum, so that their action on the vacuum creates the same states as the action of $B_i$ on the vacuum. If the operator $\mathcal{P}_\Omega$ had been a projector onto states of zero gauge-charge, it would not have projected onto the vacuum, but instead have projected onto a large subspace of the Hilbert space. However the subspace comprising all states of zero gauge-charge contains states with widely differing local-entanglement structures. So the operators $C_i$, formed by combining $Q_i$ and $\mathcal{P}_\Omega$, would not have had a large two-point function with $A_i$.

But the distinction between gravity and gauge theories is also clear from a physical point of view. Non-gravitational gauge theories contain local operators that are exactly gauge-invariant. So, in such a theory, an observer in the middle of AdS can act with a localized unitary that does not change the value of *any* observation near the boundary and is entirely invisible to the boundary observer. But this means that, in a non-gravitational gauge theory, the observer near the boundary *cannot* uniquely identify the bulk excitation.

To summarize, while in gauge theories, the Wilson lines of operators can be used to construct non-zero commutators between operators localized in distinct regions, there is no analogue of the phenomenon of complementarity, which seems to be a feature unique to quantum gravity.

Since these nonlocal relations in gravity are important in empty space, it is natural that they will also be important during black hole evaporation. As we pointed out above, the operators used in the construction of this toy model are similar to the operators used in the monogamy and cloning paradoxes.

More specifically, the operators that distil information relevant for the infalling observer from the old Hawking radiation — which are used in both the cloning and monogamy paradoxes — can be written explicitly in the form of the operators $C_i$. This is achieved by replacing $\mathcal{P}_\Omega$ in the construction above with the projector onto the black-hole microstate, and by replacing $Q_i$ from equation (11) with the appropriate operators that act on the old Hawking radiation like an operator near the horizon.

These operators are more complicated than $C_i$, but they are also fundamentally similar to $C_i$ in that they

can also be measured by measuring very high-point correlators of localized light operators in a thin shell far away from the black hole.

This strongly suggests that the monogamy and cloning paradoxes can be resolved by recognizing that even if these operators are localized far from the interior, they can nevertheless extract quantum information from the interior just as was done in the toy model above.

The toy-model also shows that this violation of naive-locality does *not* imply a breakdown of effective field theory behind the horizon — as suggested by the firewall and fuzzball proposals. This is because these violations of naive-locality also appears in empty space where effective field theory is clearly valid. There is no inconsistency between the idea that there are relations between complicated operators localized in different regions, and the fact that the geometry appears entirely smooth when probed with simple observables.

Indeed an important open problem is to precisely delineate the situations in which nonlocal effects are important. As mentioned above, these effects are clearly unimportant for questions that only reference correlators with a small number of insertions. However, this cannot be the entire story. For instance, the entanglement wedge conjecture [32] would suggest that arbitrarily complicated correlators measured inside an entanglement wedge remain meaningfully localized in the wedge and do not leak outside it. So it would be nice to devise a precise criterion that indicates when nonlocality in gravity is important.

**Acknowledgments** I am grateful to Sudip Ghosh, Monica Guica, Nima Lashkari, R. Loganayagam, Kyriakos Papadodimas, Ben Toner, Eric Verlinde, Spenta Wadia and all the members of the ICTS string group for useful discussions. I am grateful to IMSc (Chennai) and CERN (Geneva) for hospitality while this work was in progress. This work was partially supported by a Swarnajayanti fellowship of the Department of Science and Technology (India).

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
