# Peer review of "A Toy Model of the Information Paradox in Empty Space"

_SciPost Physics_

## Round 3 · Referee Report · Anonymous (Referee 1) · 2019-4-2

Strengths

  1. Addresses an important and difficult problem
  2. Interesting use of Bell correlations in QFT

Weaknesses

  1. Subtleties regarding the Hamiltonian as an operator in quantum gravity are glossed over
  2. Not all assumptions are clearly stated
  3. Some definitions should be sharpened
  4. Overall claims are stated with a strength that does not seem appropriate to the strength of the required assumptions of the argument, once these are stated in full.

Report

This manuscript aims to construct a toy model of some versions of the black hole information paradox, in Anti-de Sitter space, using states close to the vacuum.

In local QFT, operators that are localized on distinct spacelike separated regions commute, and one way to express the monogamy of entanglement is through the inequality given in Eq. (2). The idea presented in this manuscript is to consider a particular set of operators in quantum gravity that are argued to be approximately localized on distinct spacelike separated regions, and to show that these operators violate this inequality. There is no sharp contradiction, as noted at the start of the conclusion section, however the goal is to highlight issues of locality in quantum gravity, and to attempt to draw a parallel with certain statements of the black hole information paradox.

The overall idea is interesting, and addresses an important and difficult problem. However there are several points that are glossed over in the somewhat brief presentation, that undermine the strength of the claims, and that should be clarified before the paper can be considered suitable for publication.

The main such point is that the discussion of the bulk Hamiltonian, as an operator in quantum gravity, glosses over potentially important subtleties.

On p4, at the top of the second column, it is asserted that the operator $H^{can}$ (defined in the display equation) is the Hamiltonian and that it is a boundary term, citing Refs. [22]-[27]. This discussion glosses over several subtleties, most of which are well-known, and which are intricately linked to long-standing problems in quantizing gravity. The exposition appears to present at least one assumption, and/or assertion that is not solidly established, as if it were an established fact. This is very unsatisfactory. Stating clearly all assumptions will add significant value to readers who may be interested in interpretational issues raised by the overall argument of the manuscript.

Specific requested changes, including other points, are detailed in the following section.

In light of these subtleties, the claims in this manuscript may be taken with a pinch of salt by many readers; the delicate issues with the Hamiltonian as an operator in quantum gravity appear to weaken the conclusions relative to the strength with which they are presented. As a result, the question of whether this is a useful toy model for the Information Paradox is unclear to say the least.

Nevertheless the overall idea is interesting, so if the specific points described below are addressed in a way that is convincing and satisfactory, with all assumptions clearly stated, the manuscript will likely be acceptable for publication in SciPost Physics.

Requested changes

  1. On p1, towards the bottom, it is stated that "we will use the term localized to describe operators that comprise quantum fields from a region after a specific choice of gauge." However on the following page, it is argued that "complicated enough operations with localized operators" may have an effect outside the original region, at O(1). The terminology may then appear confusing (i.e. if so, was the operator ever meaningfully "localized" in the first place?). So the definitions of "localized" and "complicated" should be sharpened. In doing so it should be clarified whether the context of the definition is perturbation theory and/or free field theory, or more general than this.

  2. On p4, first column, near the bottom, it is stated that "…we can construct a bulk operator, near the boundary of the space, that projects onto the vacuum". The presentation is cavalier regarding the precise support of this operator. Here "bulk operator, near the boundary", together with both Fig. 1 and with the last paragraph of p4, suggest that the support is over a finite range of values of the radial coordinate, within the bulk. However the display equation at the top of p4 suggests that the operator is a pure boundary term, with support only on the conformal boundary of AdS. The support of the operator under consideration should be stated unambiguously.

  3. On p4, at the top of the second column, a distinction should be made between the holographic dictionary for the expectation value of the boundary CFT Hamiltonian (which is well established) and the proposed much stronger statement at the level of quantum operators in the display equation at the top of the second column of p4.

  4. The statement that the Hamiltonian is a boundary term should be sharpened and it should be made clear what version of this statement is established and what version of this is an assumption. Specifically, the statement that the Hamiltonian of classical gravity is a boundary term on-shell is established (Ref. [27]). At least a brief reference to or discussion of on-shell vs off-shell issues should be made. Moreover, the statement that the Hamiltonian, as an operator in quantum gravity in AdS, has support only near (or at) the boundary of AdS, is to my knowledge not solidly established, and there is not a consensus on this point. This subtlety should be acknowledged and this statement should be re-classified as an assumption. In addition, there is a statement in italics "$H^{can}$ is a positive operator…" which should be introduced more clearly as being an assumption (this is presently only clarified in the next paragraph).

  5. On p4 it is asserted that "...a correlator with an insertion of $P_{\Omega}$ can be calculated to arbitrary precision by combining correlators with suitably many insertions of $h_{tt}$ [5]". () This appears to be a statement about correlators in the bulk quantum gravity theory, however in Ref. [5] the primary emphasis is on correlators of CFT operators (in particular Ref. [5] deals with $P_0$ which appears to be the dual CFT analog of the $P_{\Omega}$ of this paper). It should be clarified whether the statement () is a statement about correlators in quantum gravity. If so, it is inappropriate to cite this statement to [5] and a supporting argument should be provided.

  6. On p6, starting at the end of the first column, the relation with black hole evaporation is briefly discussed. Here there is the additional subtlety that when discussing heavy states such as black holes, one cannot rely on perturbation theory around the vacuum, and this should be acknowledged.

  7. In view of the subtleties mentioned above, while the argument in this paper is interesting, there are a couple of places where the claimed implications are stated in what I consider to be inappropriately strong terms. These instances contrast with the overall appropriately conservative tone of this otherwise well-written manuscript. The following claims should be stated in a more conservative way, appropriate to the strength of the assumptions to be clarified in response to the previous points. I include suggestions in order to be clear about my meaning. (i) p2, top of column 1: "The result of this paper show, in a precise setting, that such an assumption is wrong."

  8. For example: "The results of this paper provide evidence, in a precise setting, that such an assumption may be wrong." (A rephrasing along these lines would be more in keeping with the tone of the sentence that follows this excerpt in the manuscript). (ii) p6, top of column 2: "This strongly suggests that the monogamy and cloning paradoxes can be resolved by recognizing that even if these operators are localized far from the interior, they can nevertheless extract quantum information from the interior just as was done in the toy model above. The toy-model also shows that this violation of naive-locality does not imply a breakdown of effective field theory behind the horizon..."
  9. For example: "This strongly suggests that the monogamy and cloning paradoxes might be resolved by recognizing that even if these operators are localized far from the interior, they might nevertheless extract quantum information from the interior just as was done in the toy model above. The toy-model also suggests that this violation of naive-locality may not imply a breakdown of effective field theory behind the horizon..."

If these points are addressed in a satisfactory way, the manuscript will likely be acceptable for publication in SciPost Physics.

  • validity: ok
  • significance: ok
  • originality: high
  • clarity: ok
  • formatting: excellent
  • grammar: excellent

Author:  Suvrat Raju  on 2019-04-25  [id 502]

(in reply to Report 1 on 2019-04-02)

I would like to thank the referee for the very detailed and careful report. I have implemented the changes suggested by the referee.

The referee's main point is about the operator H_{can}. I would like to emphasize that the assumption in the paper is not that this operator is the full Hamiltonian of the theory. The assumption made in the paper is weaker and it is that the "ground state of this operator has a large overlap with the true ground state of the theory." So even if H_{can} does not coincide with the true Hamiltonian for high-energy states, this does not cause any difficulty for the analysis in this paper. Stated physically, the assumption is as follows: if we measure the far-away falloff of the field and find that this far-away falloff coincides exactly with empty AdS, then we should be able to conclude that the overlap of the bulk state with empty AdS is 1 - O(1/N). Alternately stated, in the quantum theory, it should not be possible to hide energy in the bulk without at least some manifestation in the asymptotic field.

The operator H_{can} is expected to be a good quantum operator. If we have an independent definition of the bulk theory, this operator represents an integral of the quantized metric fluctuation in the asymptotic region. If the reader prefers to define the bulk theory using the CFT, rather than thinking of the bulk independently, then the operator describing metric fluctuations may be constructed by smearing the stress-tensor using an HKLL-type construction.

I would also like to address the referee's questions about the Hamiltonian being a boundary term in the bulk theory. As noted, the precise assumption required in the paper is weaker but since this point is of independent interest, it is worth addressing for readers of this correspondence. The Hamiltonian is expected to be a boundary operator in the quantum theory (not just the classical theory) provided we restrict to the set of gauge-invariant states. More precisely, diffeomorphism invariance leads to a set of first-class constraints in quantizing gravity. As a result of these constraints, the local Hamiltonian density vanishes. This represents the fact that the wave-function must be invariant under small diffeomorphisms that move bulk points forward or backward locally in time. The quantum Hilbert space is obtained by first constructing wave-functions and subsequently imposing these constraints on wave-functions. This is the content of the Wheeler-DeWitt (WDW) equation and, on any wave-function that satisfies the WDW equation, the action of the Hamiltonian is given by a boundary term.

Even though the quantum theory has ultraviolet divergences, as long as one can regulate these divergences consistent with diffeomorphism invariance, the Hamiltonian will continue to be a boundary term. This should be possible at least at all orders in perturbation theory, but if the referee knows of any specific subtleties on this front, I would be happy to include additional references pointing to these subtleties.

In my opinion, given the facts above, the assumption made in the paper is robust and well justified by what we currently know about quantum gravity in AdS. But, in any case, I have significantly expanded the discussion of the H_{can} in the paper, and stated the assumptions above clearly and readers can judge the validity of the assumption by themselves. I request the referee to examine the revised text. I think this should address the main point raised by the referee (and hopefully allow the referee to dispense with the "pinch of salt"!)

Separately, I would like to emphasize another aspect of this paper that was not directly dealt with in the referee report. One of the objectives of this paper was to bring to the fore many subtleties that are glossed over in most discussions of the information paradox. For example, many recent discussions make use of the von Neumann entropy, which is ill-defined for bulk regions in gravity that are not the entanglement wedge of some boundary region. In this paper, it is pointed out that the use of Bell correlators leads to a more precise setup. Many recent discussions of the paradox also make use of physical notions like an observer who can "collect all the Hawking radiation" and then "jump inside" the black hole. However, these verbal descriptions sidestep the entire issue of what operators must be used to measure the Hawking radiation, or how one "localizes" subsequent measurements "inside" or "outside" a region.

This toy model forces one to formalize all of these notions and presents a setting where these notions can be examined carefully (as the referee has just done!). This is why I believe that it is a useful toy model for the information paradox.

I have also implemented the other changes suggested by the referee as detailed below: 1) I have clarified the terminology. In gravity, all operators must be dressed to infinity. So, in a strict sense, no operator is confined in any finite bulk region. Thus, for instance, in the information paradox, when we speak of measurements "inside" a black hole, the operators that implement these measurements must have a global part that extends outside the black hole. Nevertheless, we would like to define some mathematical object that captures our naive notions of locality and without such an object, the cloning and monogamy paradoxes cannot even be framed. In this paper, we use gauge-fixed operators to implement our naive notion of locality. This is a common choice. (There are, of course, other possible choices such as dressing the operator to infinity by means of a geodesic but those are a little harder to implement technically.) However, if we try and push these notions too far, we are forced to confront the fact that these operators, which we thought belonged to a specific region, secretly have a global aspect.

I have inserted text in the introduction to clarify this, and I request the referee to examine this new text. In this paper, the word "localized" is used to capture our naive notion of locality and this is explicitly clarified and the reader is warned that "localized" does not mean exactly local. By "simple" vs "complicated", I meant to distinguish those operators that are low-order polynomials of elementary localized operators vs those that are not. I have now inserted text to clarify this as well.

2) Thanks. I have clarified this point. The operator is localized strictly in the asymptotic region.

3) I am using the extrapolate dictionary in its operator form, where it relates not just expectation values, but the boundary value of the bulk operator to a boundary operator. Such an operator form of the dictionary is also implicit if the bulk operators are described through an HKLL-type mapping between the boundary and the bulk. Moreover, it can be checked that in the interacting theory, multi-point correlators of the boundary values of bulk operators give correlators of the boundary operators. (See, for instance, arXiv:1104.2621.) I have inserted this discussion, stated the assumption explicitly, and also noted that it is possible that the extrapolate dictionary will be corrected at higher orders in 1/N.

4) See the discussion above. I have inserted a more detailed version of this discussion in the paper.

5) The argument here is just that the exponential can be expanded out in a high-order polynomial, which requires O[N \log N] terms. I have provided that argument now. In any calculation, it is better to just treat P_{\Omega} in the exponential form, where it is a manifestly bounded function of an asymptotic operator, and this is what we do. However, I wanted to mention this polynomial form just to bring out the similarity with the operators used in the information paradox.

6) Okay. The analogy between the paradox and black-hole evaporation is that a similar formula is expected to hold there, with P_{\Omega} -> P_{\Psi}: the projector on the black-hole microstate. However, unlike P_{\Omega}, P_{\Psi} cannot be constructed explicitly in terms of metric fluctuations, and I have noted this.

7) Okay. I have changed "such an assumption is wrong" -> "how such an assumption could fail". Also changed "can" -> "might" as suggested by the referee and "does not imply" -> "does not necessarily imply" (apart from a rephrasing of the sentence for clarity.)

I hope that this addresses all the points raised by the referee and that the paper can now proceed towards publication.

---

## Editorial Decision

resubmitted